# Validity of IMUs in Comparison to a Marker-Based-Motion Capture System for Spatio-Temporal Parameters During Wheelchair Propulsion

**DOI:** 10.3390/s25154676

**Published:** 2025-07-29

**Authors:** Lukas Karner, Lucas Schreff, Rainer Abel, Roy Müller

**Affiliations:** 1Bayreuth Center of Sport Science, University of Bayreuth, 95447 Bayreuth, Germany; 2Department of Orthopedic Surgery, Klinikum Bayreuth GmbH, 95445 Bayreuth, Germany; 3Universitätsklinikum Erlangen, Friedrich-Alexander-Universität Erlangen-Nürnberg, 91054 Erlangen, Germany

**Keywords:** spinal cord injury, wearable sensor, inertial measurement unit, motion capturing, manual wheelchair

## Abstract

Background: Manual wheelchair propulsion is often associated with pain in the upper extremities. Recording spatio-temporal parameters can optimize movement patterns and prevent injuries. This study compares a marker-based camera system with inertial measurement units to validate their use in wheelchair propulsion on a test stand. Methods: Spatio-temporal parameters of 27 manual wheelchair users propelling at three self-selected speeds (slow, normal, fast) were simultaneously recorded using a marker-based camera system and inertial measurement units, and subsequently compared between both systems. Results: A high correlation was observed among all spatio-temporal parameters (*ρ* > 0.992). The biases for the start time of hand contact with the pushrim (−0.02 ± 0.02 s), hand release from the pushrim (−0.02 ± 0.01 s), and push length (−0.45 ± 21.45 ms) were slightly overestimated, while recovery length (0.54 ± 21.02 ms), cycle speed (2.37 ± 2.67°/s), and push angle (1.75 ± 4.14°) were slightly underestimated. No bias was found for propulsion frequency. Conclusions: The spatio-temporal parameters recorded using inertial measurement units are suitable for the evaluation of manual wheelchair propulsion and can be used in a clinical context. The low acquisition costs and simple installation process may increase the use of inertial measurement units in the future.

## 1. Introduction

Using a manual wheelchair can lead to injuries in the upper extremities, including pain in the wrist and shoulder. Among manual wheelchair users, carpal tunnel syndrome occurs in 49–63% [1,2,3,4] and shoulder pain in 31–73% [5,6,7] of cases. The risk of injury and pain is increased by using inefficient propulsion patterns [8,9,10]. Following Boninger et al. [11], among the four existing stroke patterns during wheelchair propulsion, the double looping over propulsion and semicircular patterns are easy on the joints due to their low frequency. The semicircular pattern is particularly recommended [12], as its elliptical shape minimizes abrupt directional changes, thereby reducing the likelihood of injuries. The semicircular pattern also maintains a high proportion of time in the push phase over a wide range of speeds, allowing the tangential force applied to the pushrim to be distributed over a longer period, reducing force peaks and stress on the shoulder and wrist.

To record spatio-temporal parameters of movement analysis in laboratory or clinical settings, it is necessary to determine the start and end of the push and recovery phases for each propulsion cycle. Various measurement systems can be used for this purpose, including marker-based camera systems, wearable inertial sensors, and the SmartWheel. The Smartwheel is an instrumented wheel that is often used for monitoring manual wheelchair propulsion [13]. It measures forces and torques applied to the pushrim, enabling the start and end of the push phase to be detected. However, to our knowledge, the SmartWheel is no longer available as a commercial product.

With marker-based camera systems, different methodological approaches can be used to identify the propulsion phases. Hiremath et al. [14] determined the duration of push and recovery phases using a marker placed on the third metacarpophalangeal joint (3. MCP). The marker was used to detect the two changes in anteroposterior direction of the hand during each propulsion cycle. However, changes in hand direction can occur before the start of the push phase and after the end, resulting in a delay relative to the actual times. This timing is highly dependent on the propulsion patterns of the participants. Morgan et al. [15] measured the distance between the wheel axle and the 3. MCP. This distance represents approximately the radius of the wheel. A change in this radius marks the transition to the recovery phase. However, it is important to note that different individuals grip the pushrim in different ways, which may result in different distances between the wheel axle and the 3. MCP, even for the same wheel radius. This has the potential to result in an inaccurate determination of push and recovery phases. Salm et al. [16] calculated the duration of push and recovery phases using markers positioned on the wheelchair wheels, to measure their axial acceleration. The zero crossings of the acceleration data determined the start and end of push phases.

In recent years, inertial measurement units (IMUs) have become an increasingly popular alternative for analyzing manual wheelchair propulsion [17]. In several studies, the IMU sensor was attached to the hand [18,19,20]. For example, Fathian et al. [20] utilized an IMU mounted on the hand to identify the moments of hand contact and hand release during wheelchair propulsion. The technique combined a continuous wavelet transform for time-frequency analysis of acceleration data with a peak detection algorithm to determine the relevant events. Another approach [21] uses accelerometers on the upper arm, wrist and under the wheelchair seat to record the stroke number and propulsion frequency. Sensor placement under the seat showed the lowest accuracy for both parameters. Similar to Salm et al. [16], Vries et al. [22] measured the axial acceleration of the wheelchair wheels, but employed wheel-mounted IMUs instead of optical markers. Their measurements were conducted during overground propulsion outside of a laboratory setting and were directly compared to data obtained using a SmartWheel. The angular velocity measured around the wheel axis clearly shows an increasing trend during the push phase and a decreasing trend during the recovery phase. Using peak detection, the start and end of each phase can be identified with precision. The comparison revealed that the push phases measured with IMUs are found to be longer in duration compared to measurements obtained with a Smartwheel. This discrepancy can be attributed to additional accelerations of the wheelchair wheels caused by trunk movements during propulsion, which extend the detected end of the push phase and/or to the “real-life” movements (i.e., turns, acceleration and deceleration phases, as well as ascents and descents) considered in the average push duration.

A direct comparison between IMUs and marker-based motion capture systems in wheelchair propulsion analysis has not been conducted to date. This is surprising, however, given that both systems are used. To assess the extent to which IMUs offer an alternative to a marker-based motion capture system for, e.g., clinical applications, this study investigates the validity of IMUs for spatio-temporal parameters during wheelchair propulsion on a test stand, with a focus on identifying the start and end times of hand–pushrim contact. To calculate these start and end times, we use an approach similar to Vries et al. [22,23] and Salm et al. [16], where axial wheel acceleration is recorded using either IMU- or marker-based measurements. It is hypothesized that there will be no difference between the two measurement systems in detecting these events during wheelchair propulsion.

## 2. Materials and Methods

### 2.1. Subjects

A total of 27 participants were included (Table 1). Each participant used their own wheelchair. Participants were required to be 18–65 years old and able to propel a wheelchair over a distance of 100 m with a consistent movement pattern. Prior to participation, written informed consent was obtained from all subjects. This study was approved by the ethics review board of the Friedrich-Alexander University Erlangen-Nürnberg, Germany (23-20-B; 14 February 2023) and was in accordance with the Declaration of Helsinki.

### 2.2. Protocol

This study was conducted at Klinikum Bayreuth GmbH, Bayreuth, Germany. Prior to data collection, all participants were informed of the test setup and experimental procedure. The wheelchair was securely mounted on a test stand and fixed in place to prevent any unintentional movement (Figure 1). Participants were given a short familiarization period to adapt to the test setup before data collection started. The recording phase consisted of ten propulsion cycles at three different self-selected speeds: normal, fast, and slow. Movement data were captured using a camera system and IMUs.

### 2.3. Data Collection

A marker-based camera system (VICON, Oxford, UK) was used to monitor the movement of the wheelchair wheels. Three reflective markers were attached to each wheel (left and right). One marker was positioned on the wheel axle and two additional markers were mounted on the spokes using fixation plates (Figure 1). The system recorded marker positions using ten cameras at a sampling rate of 200 Hz. The accuracy of the camera system was assessed using Vicon Nexus calibration metrics. The mean world error, defined as the average deviation between reconstructed and expected 3D marker positions, was 0.30 mm (range across the ten cameras: 0.15–0.51 mm). The mean image error, defined as the average deviation between detected and predicted marker locations in the camera image, was 0.09 pixels (range: 0.07–0.14 pixels). These results indicate high calibration quality and spatial accuracy of the motion capture system.

To assess the rotational dynamics of the wheels, IMUs (Cometa, Bareggio, Italy) were attached to the spokes of both the left and right wheels using fixation plates (Figure 1). These IMUs then record the rotation around the wheel axes. The propulsion force exerted by the subject on the pushrim can be decomposed into a radial and a tangential force. The tangential force contributes to the forward motion of the wheelchair [24], thereby generating angular velocity around the wheelchair’s axle. The angular velocity profile was recorded in degrees per second, with the sampling rate of 142 Hz.

### 2.4. Data Processing

The raw data of the marker-based camera system and the IMUs were filtered using a second-order Butterworth filter with a cutoff frequency of 6 Hz [25]. The data processing was conducted using Python (version 3.13.4) within the Integrated Development Environment PyCharm (JETBRAINS, Prague, Czech Republic). The angular acceleration of the wheels was determined by calculating the angle (Θ) based on one vector *a_(t)_* rotating around the wheel axis (Figure 1) and one positioned vertical reference vector *a_ref_* [16]. This calculation was performed using Equation (1). Zero crossings of the angular acceleration were then used to identify the start times of hand contact (*t_contact_*) with and hand release (*t_release_*) from the pushrim. To ensure precise tracking, two markers were placed on the spokes of each wheel, although only one was required. An additional marker was positioned to compensate for potential occlusion caused by hand movements.(1)$$\Theta =\cos^{-1}\frac{{\vec{a}}_{\left(t\right)}\cdot {\vec{a}}_{\mathrm{ref}}}{\left|{\vec{a}}_{\left(t\right)}\right|\cdot \left|{\vec{a}}_{\mathrm{ref}}\right|}$$

The gyroscope data of the IMUs (Figure 2) were processed based on an approach proposed by Vries et al. [22]. The local minimum corresponds to *t_contact_*. Following this, the angular velocity increases, with the local maximum indicating *t_release_* and the start of the recovery phase. Based on *t_contact_* and *t_release_* various spatio-temporal parameters were calculated (Table 2).

### 2.5. Statistical Analysis

Statistical analyses were performed using the PyCharm (JETBRAINS, Prague, Czech Republic) development environment. The characteristics of the subjects were described using the mean and standard deviation (SD). Various statistical methods were applied to compare the measurement systems. The Shapiro–Wilk test was used to assess the normal distribution of all propulsion parameters. The Spearman correlation was calculated to analyze the agreement between the measurements. In addition, bias (mean difference between measurement systems) and absolute relative error were determined. To visualize the agreement between the measurement systems, Bland–Altman plots were generated showing the mean difference and the confidence interval (95%) of the differences.

## 3. Results

The accuracy of the parameter obtained from the two measurement systems is shown in Table 3. From the perspective of the IMUs, the parameters *t_contact_* and *t_release_* were systematically overestimated, with a bias of −0.02 s. The push length was also overestimated with a bias of −0.45 ms. The biases of the parameter’s recovery length (0.54 ms), cycle speed (2.37°/s), and push angle (1.75°) were underestimated. Propulsion frequency showed no systematic deviation between the two measurement systems. The largest absolute relative errors occurred for the push length (4.05%), recovery length (2.27%), and push angle (3.49%). In accordance with this, these propulsion parameters also showed lower correlations compared to the other recorded parameters (push length *ρ* = 0.994, *p* < 0.001; recovery length *ρ* = 0.998, *p* < 0.001; push angle *ρ* = 0.992, *p* < 0.001).

Figure 3 illustrates the agreement between the two systems for various propulsion speeds. The propulsion parameters show a dependency on different speeds. As the velocity increases, the push length (*R*^2^ = 0.402), cycle speed (*R*^2^ = 0.422), propulsion frequency (*R*^2^ = 0.000), and push angle (*R*^2^ = 0.277) increase, while the recovery length (*R*^2^ = 0.254) decreases.

## 4. Discussion

The aim of this study was to validate the measurement of the IMUs using the marker-based camera system. In general, the spatio-temporal parameters as determined by the IMU system showed a high degree of agreement with the reference system. However, systematic deviations were observed in some measured variables. For instance, the IMU system slightly overestimated *t_contact_* and *t_release_*. The push length was overestimated the most and had the highest absolute relative error, followed by the push angle, which was underestimated. However, the recovery length was underestimated, resulting in the third largest deviation value. The cycle speed was slightly underestimated, while no deviation between the two measurement systems could be determined for the propulsion frequency. Despite these minor deviations, the very high correlation coefficients (Table 3) across all parameters indicate that both systems capture the same underlying spatio-temporal characteristics of wheelchair propulsion with a high degree of consistency.

Wheel-mounted IMUs were compared in a recent study with data obtained from a SmartWheel [22]. However, Vries et al. [22] focused on distance covered by the wheelchair, linear velocity of the wheelchair, and number and duration of pushes. They observed a significantly different duration of the push phase whereas the push durations derived from the IMU data were longer than those from the SmartWheel data. Vries et al. [22] attributed these large differences to the trunk and arm movements during the pushing processes. A resulting acceleration of the wheelchair can occur after the push phase has been completed, which can lead to a delay of *t_contact_* and *t_release_* [26].

In clinical settings, movement analyses for wheelchair users are often performed on test stands (Figure 1). One advantage of such test stands is that they allow manual wheelchair users to perform steady, repeatable propulsion in a small space over extended periods of time. When measurements are taken on a test stand, accelerations of the wheels caused by trunk and arm movements, as observed in the overground study by Vries et al. [23], cannot occur. In this setup IMUs could present an attractive alternative to marker-based motion capture systems. While optical systems are considered the gold standard for kinematic analysis, they require specialized laboratory environments, extensive setup time, and expert operation, making them costly and labor-intensive. Moreover, they are prone to occlusion issues, which can lead to data loss and time-consuming post-processing. Outside of a laboratory setting, the use of wheel-mounted IMUs is not yet entirely clear (see comments above). However, an improvement in analytics outside the laboratory could lie in Fathian et al.’s calculation method [20]. They identify the moments of hand contact and hand release during wheelchair propulsion utilizing an IMU mounted on the hand. Their technique combined a continuous wavelet transform for time-frequency analysis of acceleration data with a peak detection algorithm to determine the relevant events. In addition to IMUs, alternative sensor systems, such as pressure sensors attached to the wheelchair user’s middle fingers, have also been explored to detect contact time with the pushrim during propulsion [27] and could be used for overground studies.

A comparison of the camera system and the IMU has already been carried out several times while walking [28,29]. In the present study the parameter cycle speed is influenced by the propulsion speed and demonstrates a comparable behavior to the results of Zahn et al. [28], who observed a stronger deviation from the reference system at higher gait speeds. As the propulsion speeds of the wheelchair users increases (from slow to fast), the deviation between the two measurement systems increases, as illustrated by the Bland–Altman plots. The duration of the gait phases behaved in opposite ways depending on the gait speed. As the speed increased, IMUs recorded a decreased stance phase and an increased swing phase. This discrepancy can be attributed to differences in the identification of the beginning of the stance phase in the two systems. In this present study, the push phases are systematically underestimated and the recovery phases overestimated with increasing propulsion speed. One possible cause is the algorithm for recognizing *t_contact_* and *t_release_* based on the IMU data. At lower speeds, the wheelchair wheel nearly comes to a standstill at the end of the recovery phase, making it difficult to identify the start of the push phase based on the minimum in the speed curve. Instead, the phase transition is recognized after a predefined threshold value is exceeded.

Some limitations of this present study require consideration. First, the sample consisted mostly of male test subjects. A more balanced gender distribution might be advantageous to extend the measurement system in terms of its applicability and validity to different user groups. However, as our measurement system focuses on mechanical parameters of the wheelchair wheels (rather than user-related kinematic or physiological data) we believe that the influence of gender on the current results is limited. Secondly, some of the participants pushed their wheelchairs faster than average (500–600°/s, see Figure 3). Since these high speeds influence the regression analysis, but the speeds themselves are not clinically relevant, this should be taken into account when using the sensors. Thirdly, this study is limited by its exclusive use of a stationary test stand, which may not reflect real-world overground propulsion conditions where trunk movement introduces noise.

## 5. Conclusions

This present study demonstrates the feasibility of using IMUs to analyze wheelchair propulsion in manual wheelchair users on a test stand, with potential for clinical application. Unlike marker-based motion capture systems, IMUs require minimal preparation and are less cost-intensive. In contrast to the specialized SmartWheel, IMUs are commercially available and more accessible, allowing for rapid and flexible data collection.

## Figures and Tables

**Figure 1 sensors-25-04676-f001:**
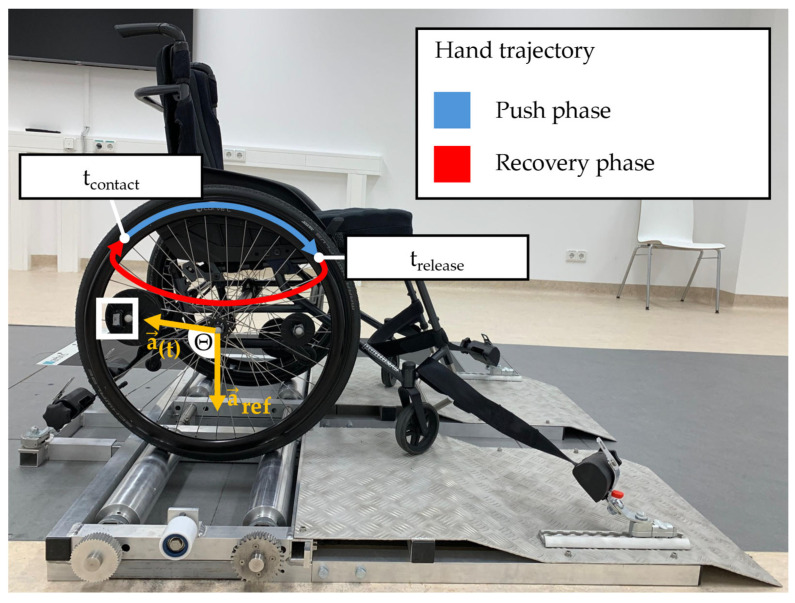
Detailed representation of the measurement systems: wheelchair on a test stand in a laboratory setting including 10 cameras, showing markers attached to the spokes and axle and an IMU attached to the spokes (white rectangle); the vectors *a_(t_*_)_ and *a_ref_* (marked in orange) are used to determine the angular acceleration based on the calculated angle *Θ*. Note: only one of the spoke markers is required to calculate *Θ*; the second marker is used solely for gap filling when occlusions occur due to hand movement.

**Figure 2 sensors-25-04676-f002:**
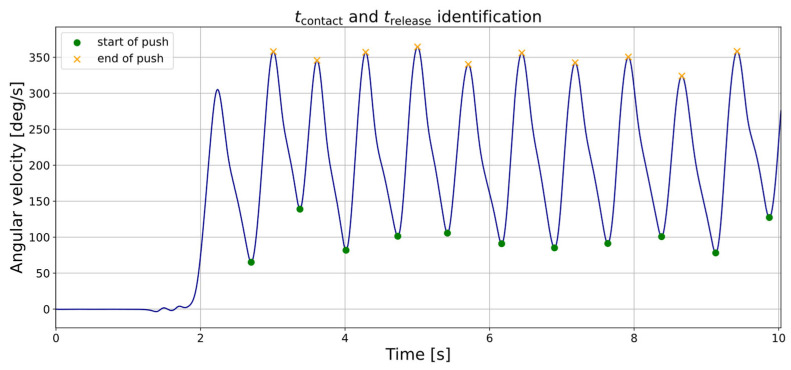
Example of angular velocity (10 push phases) measured with an IMU mounted on the spokes of the right wheel, marked in green for *t_contact_* (minima) and orange for *t_release_* (maxima).

**Figure 3 sensors-25-04676-f003:**
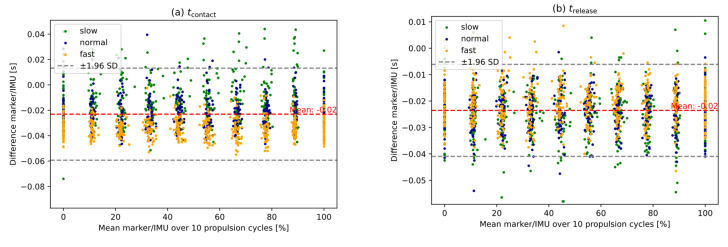
Bland–Altman plots visualize spatio-temporal parameters of wheelchair propulsion ((**a**) *t_contact_*, (**b**) *t_release_*, (**c**) push length, (**d**) recovery length, (**e**) cycle speed, (**f**) propulsion frequency, (**g**) push angle), showing the mean difference (red dashed line), regression line (black dashed line), and the confidence interval with an agreement of 95 % (between the two gray dashed lines) for two systems (marker-based camera system/IMU) for three different speeds (slow, normal, fast).

**Table 1 sensors-25-04676-t001:** Participants’ characteristics.

	Mean ± SD
Age [years]	46.78 ± 14.98
Sex [f/m]	2/25
Weight [kg]	82.37 ± 19.08
Arm length [cm]	55.48 ± 4.25
Duration of wheelchair dependency [years]	16.10 ± 13.90

**Table 2 sensors-25-04676-t002:** Detailed description of measured spatio-temporal parameters in wheelchair propulsion.

Parameter	Description
*t_contact_*	Point in time of initial hand contact with pushrim
*t_release_*	Point in time of hand release from pushrim
Push length	Duration of hand contact during the push phase
Recovery length	Duration between hand release and initial hand contact
Cycle speed	Average speed of one propulsion cycle
Propulsion frequency	Number of propulsion cycles per second
Push angle	Angle between hand contact and release on the pushrim

**Table 3 sensors-25-04676-t003:** Comparison of spatio-temporal parameters of wheelchair propulsion measured with a marker-based camera system and IMUs.

Parameter	Marker Value ^1^	IMUs Value ^1^	Bias ± SD	Abs. Error (%) ^2^	*ρ* ^3^
*t_contact_* (s)	- ^4^	- ^4^	−0.02 ± 0.02	0.36	1.000
*t_release_* (s)	- ^4^	- ^4^	−0.02 ± 0.01	0.28	1.000
Push length (ms)	415.50 ± 142.93	415.96 ± 156.51	−0.45 ± 21.45	4.05	0.994
Recovery length (ms)	739.24 ± 289.31	738.70 ± 278.72	0.54 ± 21.02	2.27	0.998
Cycle speed (°/s)	172.31 ± 121.16	169.94 ± 119.42	2.37 ± 2.67	1.76	1.000
Propulsion frequency (Hz)	0.99 ± 0.37	0.99 ± 0.37	0.00 ± 0.01	0.66	1.000
Push angle (°)	66.25 ± 24.91	64.50 ± 22.74	1.75 ± 4.14	3.49	0.992

^1^ Mean value ± SD; ^2^ Absolute relative error; ^3^ Spearman correlation coefficient *ρ*; ^4^ *t_contact_* and *t_release_* are points in time, no mean values provided.

## Data Availability

Data as well as the IMU signal processing algorithm will be made available upon reasonable request to the corresponding author.

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
