# Peer review of "Validity of IMUs in Comparison to a Marker-Based-Motion Capture System for Spatio-Temporal Parameters During Wheelchair Propulsion"

_sensors, 2025, doi:10.3390/s25154676_

Round 1

Reviewer 1 Report

Comments and Suggestions for Authors

The manuscript compares the effectiveness of IMUs and a marker-based motion capture system in measuring spatio-temporal parameters of wheelchair propulsion on a test platform, with a particular focus on identifying the start and end times of hand–pushrim contact. To further improve the quality of the manuscript, I offer the following suggestions:

(1) The study employs two systems—marker-based camera system and IMUs—to capture spatio-temporal parameters during wheelchair propulsion. However, prior to the experimental comparison, it would be helpful if the authors could provide an assessment of the measurement performance (e.g., accuracy, resolution, or sampling rate) of one or both systems to support the reliability and validity of the methods used.

(2) Overall, as a scientific paper, the scope of the work appears somewhat limited. The study mainly focuses on comparing the two systems to confirm the feasibility of IMU-based measurements. It is recommended that the authors expand the manuscript by including additional content—for example, exploring potential real-world or clinical applications of IMU-based measurement after confirming its effectiveness.

(3) In terms of formatting, the symbols representing variables in the equations (e.g., in Equation 1) or throughout the main text should be presented in italic font according to standard scientific writing conventions.

(4) Additionally, many of the cited references are from more than two decades ago; it is suggested that the authors include more recent studies, particularly those published within the past five years, to enhance the relevance and timeliness of the literature review.

Reviewer 2 Report

Comments and Suggestions for Authors

The authors present a validation study comparing inertial measurement units (IMUs) to a marker-based motion capture system for assessing spatio-temporal parameters during manual wheelchair propulsion on a test stand. Their findings demonstrate a high correlation across all parameters, supporting the potential of IMUs as a practical, low-cost alternative for clinical and performance-based use.

The study is clinically relevant, addressing a critical need to reduce upper extremity injuries in wheelchair users by enabling more accessible motion analysis. The methodology is rigorous, utilizing synchronized data acquisition, appropriate filtering, and robust statistical comparisons such as Bland-Altman plots and Spearman correlation. The results show strong agreement between systems, with minimal bias and absolute error. The discussion effectively contextualizes these findings within prior literature, highlighting both technical and clinical implications.

However, the study is limited by its exclusive use of a stationary test stand, which may not reflect real-world overground propulsion conditions where trunk movement introduces noise. The sample is also predominantly male, which restricts generalizability. Finally, the IMU signal processing algorithm could benefit from clearer documentation or code availability to improve reproducibility. Addressing these limitations would strengthen the paper, which otherwise offers a valuable contribution to wearable computing in rehabilitation.

Round 2

Reviewer 1 Report

Comments and Suggestions for Authors

The quanlity of this manuscript has been improved.